# Vulnerable Factors Affecting Urinary N-Methylformamide Concentration among Migrant Workers in Manufacturing Industries in Comparison with Native Workers in the Republic of Korea (2012–2019)

**DOI:** 10.3390/ijerph192013450

**Published:** 2022-10-18

**Authors:** Kyung-Eun Lee, Kayoung Park, Dong Joon Park, Sungkyun Park

**Affiliations:** Korea Occupational Safety and Health Agency, Ulsan 44429, Korea

**Keywords:** migrant workers, SMEs, N,N-methylformamide, threshold limit values

## Abstract

Background: Occupational studies on N-N-dimethylformamide (DMF) exposure among migrant workers in Korea are scarce. We determined the urine concentration of N-methylformamide (NMF) among migrant workers with DMF exposure and compared the data with those of native workers. Methods: Data were collected from Workers’ Special Health Examination and Work Environment Monitoring databases during 2014–2019. Workers aged ≥20 years were eligible to participate in the special health examination for DMF exposure. Urine concentrations of NMF were determined and compared between migrant and native workers. We also evaluated the factors affecting the difference in the urine concentration of NMF between the migrant and native workers. Multiple logistic regression was performed by adding confounders step by step. Results: Among 9259 subjects, 504 (5.2%) were migrant workers. The mean urinary concentration of NMF was 6.73 mg/L in migrant workers, which was significantly higher than that in native workers (2.06 mg/L, *p* < 0.001). The odds of a urine concentration of NMF > 30 mg/L were significantly higher in migrant workers than in native workers after adjusting for sex and age (odds ratio [OR] = 7.31, 95% confidence interval [CI] = 4.66–11.45). However, the odds between the native and migrant workers were not significantly different when fully adjusted for confounders (OR = 1.12, 95% CI = 0.65–1.94). Conclusions: The excessive exposure to DMF among migrant workers was attributed not to differences in biological characteristics but to their work environment. Workers must have awareness of the use of protective equipment and knowledge of hazardous chemicals that they may be exposed to, especially at the workplace.

## 1. Introduction

Nearly 4.4% of the total workers globally are immigrants, who leave their country of origin and move to another country in search of occupational opportunities [1,2]. The proportion of migrant workers has rapidly increased both globally and in the Republic of Korea (hereafter, Korea) to resolve manpower shortage [3]. According to the Survey on Immigrants’ Living Conditions and Labor Force conducted by the Ministry of Justice and Statistics Korea, the number of migrant workers increased by 6.0% from the previous year to 884,000 as of 2018, of which 45.6% were engaged in manufacturing [4]. Migrant workers were mainly recruited in small and high-risk companies wherein native workers avoided work; this caused frequent occupational injuries among migrant workers than among native workers [5,6,7]. It is assumed that the incidence of occupational diseases such as sub-acute intoxication, which shows inconspicuous or latent progression compared to mechanical injuries, is further underestimated among migrant workers [5,8,9].

The Korea Occupational Safety and Health Agency (KOSHA) reported that 25% of the intoxication-related accidents due to 10 chemicals at workplace occurred in migrant workers during 2005–2016 [10]. N-N-dimethylformamide (DMF), which could induce liver toxicity, was the most common chemical, accounting for 27.3% of the intoxication-related accidents (9 accidents affecting 24 employees), of which 33% involved migrant workers in the workplace [10]. Workers might be exposed to DMF when using chemicals, resins, fibers, coatings, inks, and adhesives, mostly in synthetic leather and textile industries. Many migrant workers who are engaged in manufacturing processes might have high exposure to DMFs at workplace because immigrants without language skills are relegated to dangerous and high-risk jobs [2,7]. Occupational health studies of migrant workers who are at risk of exposure to DMF are rare despite their social vulnerabilities because of the lack of information owing to deficiencies in the national monitoring system [2,11,12].

The present study aimed to determine the urinary concentration of N-methylformamide (NMF), a biological monitoring index (BEI) of DMF, among migrant workers who are at risk of exposure to DMF and compared with the data of native workers in the manufacturing industry. This study also assessed occupational factors associated with the risk of DMF intoxication, using data from occupational health examinations during 2014–2019.

## 2. Materials and Methods

### 2.1. Data Source

Data were obtained from the Workers’ Special Health Examination (WSHE) and Work Environment Monitoring (WEM) databases during 2014–2019, managed by the KOSHA. In accordance with Articles 125, 129, and 130 of the Occupational Safety and Health Act in Korea, the WSHE and WEM are mandatory health screening and exposure monitoring programs performed periodically among workers who are regularly exposed to any of the listed hazardous substances and various physical environments specified by the Ministry of Occupational and Employ (MOE). All companies in Korea have a unique identification number managed by the MOE. Using the companies’ own identification numbers granted by the MOE, the 2 databases of WSHE and WEM were linked by the exposure agent and the year of examination. The highest record of TWA at the same company and year was attached without weights to WSHE in the workers at risk of exposure. Among two or more records collected in a subject throughout 3 years, the oldest record was used in the analysis.

### 2.2. Study Design

This cross-sectional study was designed to enroll employees aged ≥20 years who were eligible to participate in the special health examination for DMF exposure during 2014–2019. However, participants (a) who had no records of the urine concentration of NMF and (b) who were employed in places where the airborne concentration of DMF had never been measured were excluded from the study. The enrolled population was classified into 2 groups based on whether the participants were migrant workers or native workers. If several records were found from the WSHE and WEM for a worker, then the earliest record was included in the analysis.

### 2.3. Measurements

We assessed the characteristics known to affect DMF exposure and its metabolism in the human body, including demographics (sex and age), companies where the subjects are employed (number of workers, types of industries), and DMF exposure factor (time weighted average (TWA), years of employment). According to the definition used by the European Union, companies with less than 5 and between 5 and 49 employees were classified as micro- and small-sized enterprises (MSEs), and companies with 50 or more employees were classified as medium-and large-sized enterprises [13]. For industry types in manufacturing, the 5 dominant industries where the migrant workers were the most hired as employees were chosen and classified as follows: rubber and plastic products, apparel, textiles, chemicals and chemical products, leather and related products, and others, defined as other manufacturing except the 5 industries. As for the workplace exposure indicators of DMF, the employees were classified into the “Low,” “Moderate,” and “High” groups based on the recorded urine concentration of NMF and TWA of DMF in the workplace. Employees with records of “0” or “ND (not detected)” or “Trace” were considered in the “Low” group. Others were classified into 2 groups, namely, “Moderate” and “High” based on the threshold limit values (TLVs, 5 ppm) and biological exposure indices (BEIs, 30 mg/L) adopted by the American Conference of Governmental Industrial Hygienists (ACGIH).

### 2.4. Statistical Analysis

Student’s *t*-test and Pearson’s chi-square test were used to compare the baseline characteristics between native and migrant workers. To compare the urine concentration of NMF by the types of manufacture industries, the generalized linear models were used in estimating the marginal means with 95% confidence intervals. The risk of exceeding a urine NMF concentration of 30 mg/L among manufacturing workers was assessed using multiple logistic regression. To evaluate the factors affecting the difference between migrant and native workers in the urinary concentration of NMF, 3 models were analyzed by adding demographics (sex, age), companies where the subjects were employed (number of workers, types of industries), and the DMF exposure factor (TWA, years of employment) as covariates. Statistical analysis was performed using SAS, version 9.4 (SAS Institute, Cary, NC, USA).

### 2.5. Ethical Considerations

The study protocol was approved by the ethical committee of the Institutional Review Board of the Occupational Safety and Health Research Institute (approval no. OSHRI-2021-HR-005). The need for informed consent was waived because of the retrospective nature of the study and the anonymity of the data after de-identification.

## 3. Results

Among the employees aged ≥20 years who were eligible to participate in the special health examination for DMF exposure during 2014–2019, 9763 subjects were included in this study. The included subjects had records of urine concentrations of NMF and TWA of DMF at their workplace. Among the included subjects, 5.2% were classified as migrant workers (n = 504) and the others as native workers (n = 9259).

Table 1 shows the various characteristics of the native and migrant workers, despite the fact that the proportion of male workers was higher than that of female workers in both groups (*p* = 0.794). The proportion of younger workers was higher in the migrant worker group. Subjects aged 40–49 and ≥50 years accounted for 22.0% and 18% among native workers, and migrant workers accounted for only 12.7% and 5.8%, respectively (*p* < 0.001). In addition, 60.6% of the migrant workers were employed in small-or medium-sized companies with less than 50 employees (n = 305), while 19.6% of the native workers (n = 1814) were employed in such a company (*p* < 0.001, Table 1). In terms of manufacturing type, more than half of the migrant workers were employed in the manufacture of rubber and plastics (24.8%), apparel (14.1%), and textile (11.5%) industries. On the contrary, native workers at risk of DMF exposure were employed the most (38.3%) in the manufacturing of chemicals and chemical products, where 11.5% of the migrant workers were employed. The rate of exposure to DMF exceeding the threshold limit values (TLVs) was significantly higher in migrant workers (21.6%, n = 109) than in native workers (6.3%, *p* < 0.001). The rate of urine concentration of NMF exceeding the recommended limit by ACGIH (30 mg/L) was significantly higher in migrant workers (5.6%, n = 28) than in native workers (2.2%, *p* < 0.001). The mean duration (in years) of employment among migrant workers was 1.61 years, which was significantly shorter than that among native workers (7.78 years). The mean concentration of NMF in the urine was 6.73 mg/L among migrant workers, which was also significantly higher than that among native workers (2.06 mg/L, *p* < 0.001, Figure 1). Both in the native and the migrant workers in the manufacture of leather and related products, the mean concentration of urine NMF were high compared with those in other manufacture industries. However, the mean difference of urine concentration of NMF between the native and the migrant workers in the manufacture of leather and related products was not significant (Figure 1).

Table 2 shows the factors that cause differences in the body absorption of DMF between the native and migrant workers. The odds of urine concentration of NMF exceeding 30 mg/L were significantly higher in the migrant worker group than in the native worker group when adjusting for sex and age (odds ratio [OR] = 7.31, 95% confidence interval [CI] = 4.66–11.45; Model 1). However, the risk among migrant workers dramatically decreased after adjusting additionally for the characteristics of companies (number of workers, type of manufacturing); however, the risk among migrant workers was significantly higher than that among native workers (OR = 1.72, 95% CI = 1.01–2.94; Model 2). The ORs were 8.50 (95% CI = 3.77–19.15) and 4.52 (95% CI = 2.87–7.13) in the MSEs compared with those in the medium- and large-sized enterprises (Model 2). The types of manufacturing products were also significantly associated with increasing the risks of excessive exposure to DMF, which was particularly high in manufacturing leather and related products (OR = 18.10, 95% CI = 9.87–33.21), apparel (OR = 3.64, 95% CI = 1.57–8.44), and textiles (OR = 3.51, 95% CI = 1.81–6.82) when compared with that in other manufacturing (Model 2). Finally, when fully adjusted for confounders including the ambient concentration of the DMF level at workplace and years of employment, the odds between the groups of the native and migrant workers were not significantly different (OR = 1.12, 95% CI = 0.65–1.94; Model 3).

## 4. Discussion

To the best of our knowledge, this study is the first to assess the exposure to DMF in migrant workers by comparing the data with those of the native workers using the TWA of DMF at the workplace and evaluate the urine NMF concentration at the national level. We found that the mean urinary concentration of NMF was significantly high among migrant workers compared with that among native workers despite the fact that the time of exposure and age of the migrant workers are much shorter and lower than those of the native workers. This finding supports the notion that migrant workers who are at risk of DMF exposure in manufacturing industries are highly vulnerable to occupational exposure to toxic materials. The difference in the urinary NMF concentration between the migrant and native workers could be attributed to the difference in biological metabolism according to ethnic specificity [14]. However, the fact that the significant difference in urinary NMF concentration was remarkably decreased after adjusting for several work environmental factors demonstrated that the main causes for the increase in NMF concentration in the urine were mostly attributed to the occupational environment in migrant workers (Table 2).

One of the attributing factors exceeding the BEIs in migrant workers is the size of the companies based on the number of employees. In this study, the number of employees working in MSEs was 3 times higher among migrant workers (60.6%) than among native workers (19.6%). The occupational environment in the MSEs tends to be unsafe and unregulated compared with that in medium- and large-sized enterprises because of limited resources, little knowledge about risk management processes, and deficiencies in organizational processes [15,16,17]. The International Labor Organization also reported that the MSEs attracted the majority of vulnerable workers at risk of exploitation, and migrant workers who face specific risks linked to cultural and language issues and access to social protection, as well as barriers to healthcare, were included in these MSEs. Therefore, national programs on capacity-building must be developed and adapted in the sector of occupational safety and health targeting MSEs, considering their heterogenic characteristics, including those for migrant workers [17].

The type of products manufactured at the workplace is another main factor affecting the level of exposure to DMF, leading to the difference between the migrant workers and the native workers. We found that a higher proportion of migrant workers was involved in the manufacturing industries known to lead to a high risk of exposure to DMF. The types of manufacturing products highly associated with increasing DMF exposure were rubber and plastics, leather and related products, wearing apparel, and textiles, which is consistent with the results of previous studies [18,19,20].

In this study, the risk of exposure to DMF exceeding the BEIs was significantly high even after adjusting for the air concentration of DMF in the industries that manufacture leather and related products compared with that in other manufacturing industries. Previous studies have also reported the discrepancy between BEIs and the air concentration of DMF because of the co-exposure to other chemicals or dermal absorption [18,19,21]. Kim (2014) explained that the greater the proportion of processes requiring manual handling of organic solvents, such as in the leather manufacturing industry, the larger may be difference between the biological exposure index of DMF and the measured air concentration due to the amount of additional exposure through dermal absorption [18]. On the other hand, the discrepancy between TWA and BEIs could also be attributed from the different levels of data unit. The highest measures of TWAs in the workplace were equally assigned to the workers, in spite that the workers exposed differently according to their types of working. Considering that the TWA of DMF was measured in two or more workers who were most exposed in the workplace at the same time, the TWA of DMF could be overestimated. However, the main result that the migrant workers were in vulnerable to DMF exposure in workplace would not be changed in terms that the ORs in the migrant workers was dramatically decreased after adjusting with the characteristics of companies in Table 2.

This study has several limitations that need to be considered when interpreting the findings. First, the WSHE and WEM databases used in this study contained secondary data collected from nationally designated institutions; residual confounding cannot be ruled out because of the lack of variables, especially information about specific jobs as well as co-exposures that affect the BEIs. Second, because the data used in this study were collected from heterogeneous institutions located across the country, the homogeneity of data might not be consistent despite KOSHA’s efforts to control quality in the WSHE and WEM. Therefore, we compared urinary NMF concentrations between the native and migrant workers twice after removing the limit of detection (LOD) value and after imputation to and the half of the minimum in the observed values (=0.0005 mg/L) for sensitivity analysis. Although the urine concentration of NMF in workers could be underestimated due to the substitution to the half of the minimally detected value, the overall results would not be changed. Because assessing whether the urine concentration exceeded 30 mg/L was rarely affected by the heterogeneity of LODs among various devices. Third, we modified the measured values to categorical scales in the adjusted regression to minimize the effect of heterogeneity. Finally, the small number in subgroup could have affected a wide range of confidence intervals and the risk of overestimation when producing ORs with fully adjustment with multiple variables (Appendix A).

## 5. Conclusions

In conclusion, the results of this study support the vulnerability of migrant workers in terms of their working environment and DMF exposure by analyzing data from a national database [8,22]. The results also showed that the excessive exposure to DMF among migrant workers was attributed not to their biological characteristics but to their work environment, including working in small-sized companies, having poor ventilation systems leading to high ambient levels of DMF, and using inappropriate personal protective equipment among highly manipulating processing workers [9]. It has been reported that DMF exposure is affected mainly by the effectiveness of local exhaust ventilation and appropriate wearing of personal protective equipment [23]. Therefore, it is necessary to develop an educational program, especially in small-sized enterprises, that considers the diversity of languages of migrant workers so that they can be advised on the proper use of personal protective equipment and provided with knowledge of the dangers of chemicals they are exposed to in their workplace.

## Figures and Tables

**Figure 1 ijerph-19-13450-f001:**
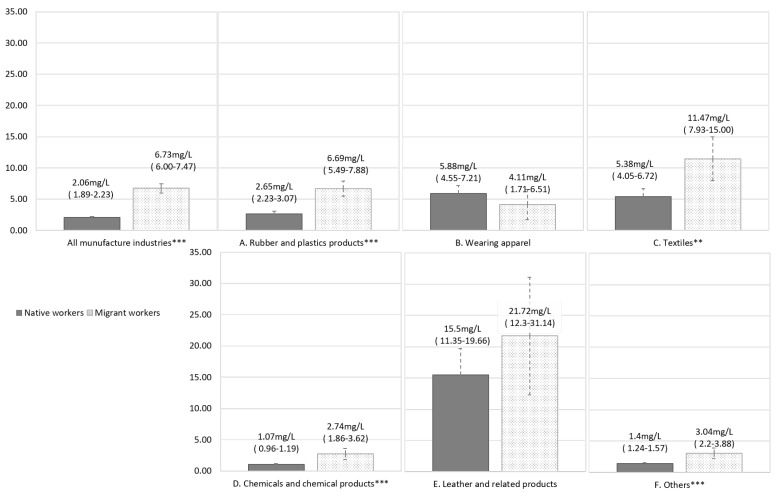
Urine NMF concentration in the native and migrant workers by the types of manufacture industries. (**A**–**F**) show the urine NMF concentrations by the native and migrant workers in the manufactures of rubber and plastics products, wearing apparels, textiles, chemicals and chemical products, leather and related products and other manufacturing industries, respectively. Estimated marginal means (95% confidence intervals) were calculated using generalized linear model. Non-detected values under LOD or those recorded as “0” were calculated by substitution with half of the minimally detected value (=0.0005 mg/dL). **, *p* < 0.01; ***, *p* < 0.001; NMF = N′-methylformamide; LOD = limit of detection.

**Table 1 ijerph-19-13450-t001:** Characteristics of native and migrant workers exposed to DMF in manufacturing industries (2012–2019).

Characteristics	Native Workers	Migrant Workers	*p* *
	(N = 9259)	(N = 504)	
**Sex, n (%)**			0.794
Male	7629 (82.4)	413 (81.9)	
Female	1630 (17.6)	91 (18.1)	
**Age group, n (%)**			<0.001
20–29 years	2868 (31.0)	214 (42.5)	
30–39 years	2681 (29.0)	197 (39.1)	
40–49 years	2041(22.0)	64 (12.7)	
≥50 years	1669 (18.0)	29 (5.8)	
**Number of workers, n (%)**			<0.001
<5	112 (1.2)	23 (4.6)	
5–50	1702 (18.4)	282 (56.0)	
≥50	7447 (80.4)	199 (39.5)	
**Types of industries/Manufacturers of, n (%)**			<0.001
Rubber and plastic products	1034 (11.2)	125 (24.8)	
Wearing apparel	230 (2.5)	71 (14.1)	
Textiles	406 (4.4)	58 (11.5)	
Chemicals and chemical products	3549 (38.3)	58 (11.5)	
Leather and related products	236 (2.5)	46 (9.1)	
Other manufacturing	3804 (41.1)	146 (29.0)	
**TWA of DMF, n (%) ^ǁ^**			<0.001
Low	5439 (58.8)	134 (26.6)	
Moderate	3233 (34.9)	261 (51.8)	
High	580 (6.3)	109 (21.6)	
**Urine concentration of NMF, n (%) ^ǁ^**			<0.001
Low	3013 (32.5)	90 (17.9)	
Moderate	6151 (66.4)	386 (76.6)	
High	95 (2.2)	28 (5.6)	
**Years of employment, years (SD)**	7.78 (9.02)	1.61 (2.06)	<0.001

* *p* values were calculated using Pearson’s chi-square and Student’s *t*-tests. **^ǁ^** Employees with records of “0” or “ND (not detected)” or “Trace” were considered in the “Low” group. Others were classified into 2 groups, namely, “Moderate” and “High” based on the threshold limit values (TLVs, 5 ppm) and biological exposure indices (BEIs, 30 mg/L) adopted by the ACGIH. DMF = Dimethylformamide; SD = Standard deviation; TWA = Time-weighted average; ND = Not detected; TLVs = Threshold limit values; ACGIH = American Conference of Governmental Industrial Hygienists.

**Table 2 ijerph-19-13450-t002:** Risk of urine concentration of N′-methylformamide > 30 mg/L among the workers.

Characteristics	Model 1			Model 2			Model 3		
	OR	(95% CI)		OR	(95% CI)		OR	(95% CI)	
**Group**									
Native workers	Reference			Reference			Reference		
Migrant workers	7.31	(4.66–11.45)		1.72	(1.01–2.94)		1.12	(0.65–1.94)	
**Sex**									
Male	0.94	(0.59–1.50)		1.52	(0.92–2.50)		1.64	(0.97–2.76)	
Female	Reference			Reference			Reference		
**Age group**									
20–29 years	Reference			Reference			Reference		
30–39 years	0.69	(0.4–1.2)		0.53	(0.30–0.94)		0.54	(0.30–0.97)	
40–49 years	1.46	(0.88–2.45)		0.80	(0.45–1.42)		0.73	(0.39–1.34)	
≥50 years	2.74	(1.70–4.42)		.89	(0.50–1.60)		1.06	(0.57–1.96)	
**Number of workers**									
<5				8.50	(3.77–19.15)		11.44	(5.00–26.18)	
5–50				4.52	(2.87–7.13)		3.39	(2.13–5.38)	
≥50				Reference			Reference		
**Types of industries/Manufacturer of**									
Rubber and plastic products				2.06	(1.07–3.99)		0.93	(0.48–1.83)	
Wearing apparel				3.64	(1.57–8.44)		1.77	(0.76–4.11)	
Textiles				3.51	(1.81–6.82)		1.71	(0.87–3.38)	
Chemicals and chemical products				0.58	(0.27–1.23)		0.88	(0.41–1.89)	
Leather and related products				18.10	(9.87–33.21)		7.32	(3.95–13.56)	
Other manufacturing				Reference			Reference		
**TWA of DMF**									
Not detected or 0							Reference		
≤5 ppm							7.65	(3.19–18.35)	
>5 ppm							36.03	(14.61–88.83)	
**Years of employment**							0.92	(0.88–0.96)	

Model 1 was adjusted for sex and age. Model 2 was adjusted for Model 1 + Characteristics of companies (the number of total workers and the type of manufacturers). Model 3 was adjusted for Model 2 + DMF exposure (TWA of DMF and years of employment). DMF = dimethylformamide; TWA = time-weighted average; OR = odds ratio; CI = confidence interval.

## Data Availability

Data was obtained from Occupational Safety and Health Institute (OSHRI), Korea Occupational Safety and Health Agency (KOSHA) and are available with the permission of OSHRI KOSHA.

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
