# Peer review of "Vulnerable Factors Affecting Urinary N-Methylformamide Concentration among Migrant Workers in Manufacturing Industries in Comparison with Native Workers in the Republic of Korea (2012–2019)"

_ijerph, 2022, doi:10.3390/ijerph192013450_

Round 1

Reviewer 1 Report

General comments

This study aimed compared the urinary concentration of N-methylformamide (NMF) between 504 migrant and 9254 native workers in the manufacturing industry, and assessed factors with impact on this difference. Data from occupational health examinations during 2014-2019 managed by the Korea Occupational Safety and Health Agency (KOSHA) was used in the study. The mean urinary concentration of NMF among migrant workers was significantly higher than in native workers. High urinary concentration was associated with the type and size of industry and with airborne exposure level to N-N-dimethylformamide (DMF).

The study lends support to the assumption that migrant workers are at increased risk of excessive chemical exposure compared to native workers. Such documentation serves important as a background information for preventive efforts to reduce and avoid social equality at workplaces. However, the manuscript has serious weaknesses that need to be considered.

Specific comments

The part dealing with NMF in the urine is unclear and confusing;

l. 140-141 & Figure 1; The mean level and (upper) 95% CI of NMF is very low compared to the biological monitoring index (BEI) of DMF (30 mg/L). Consequently, the number of workers with NMF-values> 30 mg/L is presumably low. However, these numbers, which is vital as a basis for Table 2, have not been given. In Table 2 the number of workers (migrant and native) should be given for the subgroups within all variables to improve transparency in order to better evaluate the results. Discussion on weaknesses related to any small numbers in subgroups would be relevant.

Footnote b in Figure 1 states that “Non-detected values under LOD or those recorded as “0” were calculated by substitution with half of the LOD (=0.83 mg/dL)”. How could the mean level of detected cases among native workers be 0.36 mg/dL, i.e. lower than half of the LOD? Further, Figure 1 needs a y-axis.

l.94-101; It seems from the text that the workers are grouped into low, moderate and high exposure based on both air and urine concentration, while in Table 1 only the air concentration is stated as the grouping variable. What is correct? And please describe how DMF air exposure level was assigned to each worker! Were all workers from within each company assigned the same exposure value, or were they individually based? Weaknesses related to this should be discussed.

l.123 and Table 1; The correct number of migrant workers seems to be 504, not 405.

Reviewer 2 Report

Nicely done.  This paper documents levels of urinary NMF, which reflects DMF exposure in native Korean and migrant workers in factories of various sizes that manufacture various types of goods in Korea.  The results illustrate the differences between the working conditions in small, medium and large workplaces (as reflected in higher NMF levels in smaller operations), and in exposure levels in different types of factories.  They found higher levels of NMF in the urine of migrant workers, but this difference appears to reflect their greater likelihood to work worksites with higher levels of DMF exposure.  The basic data is valuable and interesting. In addition, this work illustrates the vulnerability of migrant workers to higher levels of exposure due to the types of factories in which they are most likely to be employed. 

Author Response

Thank you for your consideration, we hope this revised manuscript will be suitable for publication in IJERPH. We appreciate for the generous review again.

Round 2

Reviewer 1 Report

The authors have responded adequately to most of my comments. The revised Table 1 now includes the number of workers with NMF-values> 30 mg/L. However, Figure 1 should be made more illustrative by showing the NMF-levels for native and migrants stratified by industry. Then Figure 1 would probably show that the Leather industry is a driver for the difference between the natives and migrants, and that this industry have higher levels than the other industries (and presumably with upper 95% CI above BEI). The revised Figure 1 should only include bars for all cases with substitution. Instead the findings related to the detected values should be given in the text.

Check whether NMF has been substituted incorrectly with DMF in lines 214-221.
